# Opposing Roles of FoxA1 and FoxA3 in Intrahepatic Cholangiocarcinoma Progression

**DOI:** 10.3390/ijms21051796

**Published:** 2020-03-05

**Authors:** Raynoo Thanan, Waleeporn Kaewlert, Chadamas Sakonsinsiri, Timpika Chaiprasert, Napat Armartmuntree, Duangkamon Muengsaen, Anchalee Techasen, Poramate Klanrit, Worachart Lert-itthiporn, Somchai Pinlaor, Chawalit Pairojkul

**Affiliations:** 1Department of Biochemistry, Faculty of Medicine, Khon Kaen University, Khon Kaen 40002, Thailand; walee@kkumail.com (W.K.); schadamas@kku.ac.th (C.S.); timpikachai@gmail.com (T.C.); napat.armart@gmail.com (N.A.); duanggamon.m@gmail.com (D.M.); porakl@kku.ac.th (P.K.); woracle@kku.ac.th (W.L.-i.); 2Cholangiocarcinoma Research Institute, Khon Kaen University, Khon Kaen 40002, Thailand; anchte@kku.ac.th (A.T.); psomec@kku.ac.th (S.P.); 3Faculty of Associated Medical Sciences, Khon Kaen University, Khon Kaen 40002, Thailand; 4Department of Parasitology, Faculty of Medicine, Khon Kaen University, Khon Kaen 40002, Thailand; 5Department of Pathology, Faculty of Medicine, Khon Kaen University, Khon Kaen 40002, Thailand; chawalit-pjk2011@hotmail.com

**Keywords:** FoxAs, FoxA1, FoxA2, FoxA3, cancer, cholangiocarcinoma

## Abstract

Cholangiocarcinoma (CCA), a malignancy of biliary epithelium, is related to liver stem cell deregulation. FoxAs are a group of transcription factors that play critical roles in liver stem cell differentiation. In this study, the expression levels of FoxAs (i.e., FoxA1, FoxA2 and FoxA3) were detected in intrahepatic CCA tissues and the functions of FoxAs were studied in CCA cell lines. FoxA1 and FoxA2 were mainly localized in the nuclei of normal bile duct (NBD) cells and some of the cancer cells. Low expression of FoxA1 in CCA tissues (72%) was significantly correlated with poor prognosis. FoxA3 expression of CCA cells was localized in the nucleus and cytoplasm, whereas it was slightly detected in NBDs. High expression of FoxA3 in cancer tissues (61%) was significantly related to high metastasis status. These findings suggest the opposing roles of FoxA1 and FoxA3 in CCA. Moreover, the FoxA1-over-expressing CCA cell line exhibited a significant reduction in proliferative and invasive activities compared to control cells. Knockdown of FoxA3 in CCA cells resulted in a significant decrease in proliferative and invasive activities compared with control cells. Taken together, in CCA, FoxA1 is down-regulated and has tumor suppressive roles, whereas FoxA3 is up-regulated and has oncogenic roles.

## 1. Introduction

The forkhead box (Fox) proteins are the transcription factors which share homology in their forkhead (winged helix) DNA binding domain. The Fox family, which contains more than 50 members, plays important roles in cell proliferation, cell cycle, apoptosis and stem cell differentiation [1,2]. The Fox family subfamily A (FoxAs) consists of three isotypes: FoxA1/hepatocyte nuclear factor (HNF)3α, FoxA2/HNF3β and FoxA3/HNF3γ. The forkhead box of FoxA1, FoxA2 and FoxA3 resembles the structure of the linker histone H1 and plays major roles in chromatin remodeling [3]. Their functions have been identified as “pioneer factors” that bind to promoters and enhancers enable chromatin access for other tissue-specific transcription factors. Thus, they play important roles in multiple stages of mammalian life, beginning with the early development, continuing during organogenesis, and finally in the metabolism and homeostasis of the adult stage [4]. FoxAs were found to induce liver stem cells differentiation by opening compacted chromatin structures within liver-specific target genes such as a hepatoblast marker alpha-fetoprotein, albumin and transthyretin [5,6]. Liver stem cells are defined to the liver stem/progenitor cells that have ability to differentiate into hepatocytes and cholangiocytes, which are also called bi-potential liver stem cells or hepatic stem cells [7].

Cholangiocarcinoma (CCA) is a liver cancer that shares many phenotypes of cholangiocytes and the hepatic stem cells [8]. The etiology of CCA is a chronic inflammatory disease of the bile duct [9]. In Thailand, liver fluke (*Opisthrochis viverrini*) infection is a major risk factor for CCA [10]. Because the parasites lodge in the intrahepatic bile ducts, CCA in Thailand is almost exclusively an intrahepatic type. Chronic inflammation is due mainly to the parasite infection and often leads to a wide range of processes, e.g., oxidative stress, tissue injuries and fibrosis. Activation of hepatic stem cells by liver injury is necessary for liver regeneration [11]. However, if this repairing process occurs under oxidative stress, it may disturb the stem cell differentiation and consequently contribute to CCA carcinogenesis. Prolonged exposure to oxidative stress is known to involve in tumorigenesis and expression of stem-like cell properties of the immortal cholangiocytes [12,13]. In addition, CCA cells have liver stem/progenitor cells phenotypes. For example, CCA cells express cytokeratin 19 (cholangiocyte marker) and albumin (hepatocyte marker) and, at the same time, express several stem cell markers (oval marker 6, CD133 and Oct3/4) [8]. These previous findings suggest that CCA genesis is related to deregulation of hepatic stem cells, which may be associated with the alterations of FoxAs expressions and activities.

FoxA1 protein expression was increased in relation to the activation of estrogen/androgen receptors in breast cancer cell progression [14]. FoxA2 expression was related with cell proliferation, migration and epithelial mesenchymal transition (EMT) processes of colon cancer [15]. High expressions of FoxA1 and FoxA3 in lung cancer were related with poor overall survival, whereas FoxA2 overexpression was associated with a better overall survival, suggesting the opposing roles between FoxA1/A3 and FoxA2 in the carcinogenesis [16]. FoxA1, FoxA2 and FoxA3 play synergistic roles in metastasis of esophageal cancer [17]. Expressions of estrogen receptors and EMT markers are induced along with CCA progression [18,19]. Moreover, the known FoxAs targeting genes such as albumin and alpha-fetoprotein were highly expressed in CCA [8,20,21]. However, the expressions and roles of FoxA1, FoxA2 and FoxA3 in CCA are not yet investigated. We hypothesized that aberrant expression and function of FoxAs are involved in CCA progression via induction of stem-like cell and tumorigenic properties of the cancer cells. In this study, the expression patterns of FoxAs in intrahepatic CCA tissues (*n* = 74) were examined using immunohistochemistry. Then, the expression patterns were analyzed with clinical data including age, sex, metastasis status and survival rates of the CCA patients. Subsequently, the two FoxA isoforms (FoxA1 and FoxA3) which were associated with the aggressive clinical data such as poor prognosis and higher metastasis were selected for the functional analyses using CCA cell lines. The stem-like cell properties were investigated using spheroid formation and expressions of liver stem cell markers. The tumorigenic properties were measured using cell proliferation and cell invasion assays.

## 2. Results

### 2.1. Expression of FoxAs in Intrahepatic CCA Tissues

Expression levels of FoxAs (light brown to dark brown) in intrahepatic CCA tissues are shown in Figure 1. FoxAs were weakly detected in normal hepatocytes located adjacent to the tumor areas. FoxA1 was highly detected in the nucleus and cytoplasm of normal bile duct (NBD) epithelial cells adjacent to tumor tissues and some CCA cells in the tumor tissues, as shown in Figure 1 (top). However, in most CCA tissues (72%; 53/74), low expression of FoxA1 was observed. FoxA2 was detected in the nucleus and cytoplasm of some cancer cells in the tumor tissues, as shown in Figure 1 (middle). Moreover, 80% (59/74) of CCA tissues had low FoxA2 expression pattern. FoxA3 was highly detected in the nucleus and cytoplasm of the cancer cells. However, it was weakly detected in the NBD cells in tumor areas. The majority of CCA tissues (61%; 45/74) had high level of FoxA3 expression.

### 2.2. Correlations of FoxAs Expression in Intrahepatic CCA Tissues and Clinicopathological Data

The correlations of FoxAs expressions in the CCA tissues and clinicopathological data of the patients are shown in Table 1. The correlation between FoxAs expressions and the survival time of CCA patients were analyzed and the results are presented in Figure 2. Low FoxA1 expression was significantly correlated with poor prognosis. Moreover, the Cox proportional-hazards model indicated that FoxA1 expression levels was an independent factor involved in the overall survival rate (*p* = 0.031, adjusted hazard ratio = 1:0.558 and 95% CI = 0.326–0.953). The median survival of CCA patients with high FoxA1 expression was 364 days and the median survival of CCA patients with low FoxA1 expression was 256 days. FoxA2 expression did not correlate with any clinical data of CCA patients. Moreover, high expression of FoxA3 in CCA tissues was significantly correlated with metastasis status and patient’s ages.

According to the data analysis, low FoxA1 expression was related with short survival rates and high FoxA3 expression was correlated with metastasis status of CCA patients. These results suggest that FoxA1 may have tumor suppressive roles, whereas FoxA3 might be involved in the induction of CCA progression. Therefore, FoxA1 and FoxA3 were selected for further functional analyses using CCA cell lines.

### 2.3. Pan-Cancer Gene Expression Analysis

We searched further for the expressions of the FoxAs in other cancer types. Gene expression data of 33 tumor types from The Cancer Genome Atlas (TCGA) database was analyzed through Gene Expression Profiling Interactive Analysis (GEPIA) (Appendix A) [22,23]. While FoxA1 was low expressed in kidney chromophobe and uveal melanoma, the expression was higher in prostate adenocarcinoma and breast invasive carcinoma when compared to cholangiocarcinoma (Appendix A). The expressions of FoxA2 and FoxA3 in cholangiocarcinoma were at similar levels with those in colon adenocarcinoma, liver hepatocellular carcinoma, pancreatic adenocarcinoma, rectum adenocarcinoma, and stomach adenocarcinoma (Appendix A). 

### 2.4. Functional Analysis of CCA Cell Lines Related to FoxA1 and FoxA3 Baseline Expression Levels

The mRNA and protein expression levels of FoxA1 and FoxA3 in KKU-100 and KKU-213 CCA cell lines were measured using real-time PCR and immunocytochemical analysis. As shown in Figure 3A–D, KKU-213 cell line had low FoxA1 expression and high FoxA3 expression compared to KKU-100 cell line. Moreover, KKU-213 cells have significantly higher cell proliferation (Figure 3E) and invasion (Figure 3F,H) activities compared to KKU-100 cells. Notably, KKU-213 could form spheroid in hanging drop culture, whereas KKU-100 failed to form spheroid under the same condition (Figure 3G). The stem cell marker (CD133 and Oct3/4) expression levels were also significantly increased in KKU-213 cells compared to KKU-100 cells, as shown in Figure 3I,J, respectively. These results indicate that FoxA1 and FoxA3 baseline expressions might be involved in the progression of CCA cells, including cell proliferation, invasion, and stem cell properties.

### 2.5. Roles of FoxA1 in CCA Cell Line

KKU-213, which has low FoxA1 expression level, was selected for FoxA1 overexpression experiments. KKU-213 cell line was transfected with the FoxA1 expression vector (FoxA1 vector) and the control vector (empty vector). FoxA1 mRNA and protein expression levels were significantly increased in the FoxA1-overexpressing cells compared to the control cells (Figure 4A,B). Then, the cell proliferation and invasion activities were measured. Cell proliferation and invasion activities of KKU-213 cells were significantly reduced after overexpression of FoxA1 (Figure 4C–E). However, FoxA1-overexpression did not affect the spheroid formation and stem cell marker expressions (CD133 and Oct3/4) compared to the control cells (data not shown). These results indicate that FoxA1 plays tumor suppressive roles in the CCA cell line via the repressions of cell proliferation and invasion activities.

### 2.6. Roles of FoxA3 in CCA Cell Line

Since KKU-213 had high FoxA3 expression level, again, this cell line was used for FoxA3-knockdown experiments. KKU-213 cell line was transfected with either the specific siRNA for FoxA3 (siFoxA3) or the control siRNA (scramble). After that, the cell proliferation and invasion activities were measured. FoxA3 mRNA and protein expression levels were significantly decreased in the FoxA3-silencing cells compared to the control cells (Figure 5A,B). Cell proliferation and invasion activities of KKU-213 cells were significantly reduced after FoxA3-silencing (Figure 5C–E). However, the FoxA3-silencing did not affect spheroid formation and the stem cell marker expressions compared to the control cells (data not shown). These results indicate that FoxA3 plays significant roles in CCA progression via inductions of cell proliferation and invasion activities.

## 3. Discussion

The expression of FoxA1 was reduced in intrahepatic CCA cells compared to NBD cells located at the tumor adjacent areas. Intrahepatic CCA patients who had low FoxA1 expression level in the tumor tissues showed significantly shorter survival rate. When cellular activities of KKU-100 cell line that has high FoxA1 expression and of KKU-213 cells that had low FoxA1 expression were compared, the former had significantly slower cell proliferation and lower invasion and spheroid formation activities compared to the latter. These results suggest that FoxA1 has tumor suppressive roles in CCA and its down-regulation may be associated with CCA progression. Accordingly, the functions of FoxA1 in CCA cell progression were investigated using FoxA1 overexpression in a CCA cell line, KKU-213. The results show that cell proliferation and cell invasion activities of FoxA1-overexpressing CCA cells were significantly reduced compared to the control cells. Since spheroid formation and stem cell marker expressions of KKU-213 cell line were not affected by FoxA1 overexpression, FoxA1 protein may not affect the stem cell properties of CCA cells. Taking these results together, FoxA1 may exert tumor suppressive roles in CCA via the inhibition of cell proliferation and invasion activities.

Tumor suppressive roles of FoxA1 were also reported in liver cancer [24] and pancreatic cancer [25]. On the other hand, FoxA1 was reported to act as oncogene in acute myeloid leukemia [26], breast cancer [27], prostate cancer [28], esophageal adenocarcinoma [29], and lung adenocarcinomas [29]. Focusing on the tumor suppressive roles, FoxA1 inhibited liver cancer cell invasion via up-regulation of microRNA-122 (miR-122) [24]. In addition, FoxA1 inhibited the epithelial-to-mesenchymal transition (EMT) in pancreatic ductal adenocarcinoma [25]. The miR-122 was reported to reduce CCA cell proliferation and invasion activities [30]. EMT process was clearly demonstrated to be involved in CCA cell invasion and proliferation activities as well as aggressive clinical outcomes of CCA patients such as poor prognosis [19]. From the literature review and the present results, we can conclude that FoxA1 acts as tumor suppressor gene in CCA via inhibition of cancer cell proliferation and invasion activities and down-regulation of FoxA1 in CCA may subsequently reduce miR-122 expression and induce EMT process leading to intrahepatic CCA progression with poor prognosis.

In this study, FoxA3 was highly expressed in cytoplasm and nucleus of intrahepatic CCA cells compared to NBD cells. High FoxA3 expression was significantly related with positive metastasis status and age of CCA patient. Additionally, high FoxA3 expression CCA cell line (KKU-213) was increased in cell proliferation rate, cell invasion activity, and stem-like cell properties compared to the low FoxA3 expression CCA cell line (KKU-100). These indicated that FoxA3 may be associated with oncogene property in CCA. The function of FoxA3 in relation to CCA progression was further studied using specific siRNA for FoxA3-knockdown experiment in KKU-213 cell line. The results indicate that FoxA3-silencing KKU-213 cells had significantly reduced cell proliferation and invasion rates compared to the control cells. However, FoxA3 expression in CCA cell line may not affect the stem cell properties as it has no effect on spheroid formation and stem cell marker expressions. Therefore, this confirmed that FoxA3 has oncogenic functions via the inductions of proliferation and invasion of the cancer cells lead to intrahepatic CCA progression with aggressive clinical outcomes such as high metastasis.

FoxA3 protein expression is age-dependent and may be relevant in diminishment of the thermogenic capacity of fat tissues during the aging process [31]. Transfections of HNF4A, HNF1A and FoxA3 suppressed hepatocellular carcinoma (HCC) cell proliferation, suggesting tumor suppressive roles of FoxA3 in HCC [32]. High expression of FoxA3 was correlated with poor prognosis in lung cancer [16]. FoxA3 was also up-regulated and promoted metastasis in esophageal cancer cells through the regulations of FoxA1 and FoxA2 expressions [17]. These suggested that FoxA3 plays oncogenic function in lung cancer and esophageal cancer via elevation of FoxA1 and FoxA2 expressions. However, in this study, FoxA1 and FoxA2 expression levels were not changed in siFoxA3-treated KKU-213 cells compared to the control cells (data not shown). Therefore, the molecular mechanisms of FoxA3 to induce CCA progression should be investigated in future.

## 4. Materials and Methods

### 4.1. Intrahepatic CCA Tissues

Seventy-four paraffin-embedded human intrahepatic CCA tissues were obtained from the specimen bank of the Cholangiocarcinoma Research Institute, Khon Kaen University, Khon Kaen, Thailand. The protocols of collection and study plan were approved by the Ethic Committee for Human Research, Khon Kaen University (#HE571283 date of approval 9 March 2018 and #HE611577 date of approval 18 December 2018).

### 4.2. Immunohistochemistry

The expressions and localizations of FoxA1, FoxA2 and FoxA3 in the CCA tissues were determined by immunohistochemistry using specific primary antibodies: mouse monoclonal anti-FoxA1 (ab55178, Abcam, Cambridge, UK), rabbit monoclonal anti-FoxA2 (ab108422, Abcam, Cambridge, UK), or rabbit polyclonal anti-FoxA3 (sc25357, Santa Cruz Biotechnology, Santa Cruz, CA, USA) antibodies. Briefly, the paraffin-embedded tissues were de-paraffinized in xylene and rehydrated through descending series of ethanol. Antigen retrieval was performed using pressure cooking in 10 mM sodium citrate buffer (pH 6) for 5 min. Endogenous hydrogen peroxidase activity and non-specific binding were blocked by 0.3% (v/v) hydrogen peroxide in phosphate buffered saline (PBS) and 10% skim milk in PBS, respectively. The sections were incubated with primary antibody at 4 °C overnight and then incubated with peroxidase-conjugated Envision™ secondary antibody (DAKO, Glostrup, Denmark) at room temperature for 1 h. Peroxidase activity was developed by 3, 3′-diaminobenzidine tetrahydrochloride (DAB) substrate kit (Vector, Laboratories, Inc., Burlingame, CA, USA). The sections were counterstained with Mayer’s hematoxylin and dehydrated with the ascending ethanol series followed by mounting with mounting solution.

The immunostaining index (IHC score) ranged from 0 to 12 and was scored by multiplying of the intensity and frequency of DAB-staining results (the intensity scored from 1 for weak signal (light brown) to 3 for strong signal (dark brown) and the frequency categorized as 0 = none, 1+ = 1–25%, 2+ = 26–50% and 3+ = 51–75% and 4+ ≥ 75%). The IHC score was categorized as high (>4.0) and low (≤4.0).

### 4.3. Pan-Cancer Gene Expression Analysis

Gene expression data of FoxAs were analyzed using GEPIA [23], a web-based interactive tool for analyzing expression data. All 33 cancer types from The Cancer Genome Atlas (TCGA) database were selected for the analysis (Appendix A). The RNA sequencing expression profiles for the genes were determined by log2 (Transcripts per million (TPM) + 1) transformed expression data.

### 4.4. Cell Lines

Human CCA cell lines, KKU-100 and KKU-213, were established from primary tumors of CCA patients in the Srinagarind Hospital, Khon Kaen University, Thailand [33]. All cell lines were maintained in the Cholangiocarcinoma Research Institute, Khon Kaen University, Khon Kaen, Thailand. The CCA cell lines were established from intrahepatic CCA tissues and also stocked at the Japanese Collection of Research Bioresources (JCRB) Cell Bank (JCRB1568; KKU-100 and JCRB1557; KKU-213). The cell lines were cultured in Ham’s F-12 complete medium that composed of Ham’s F-12 (Gibco^®^, Life Technologies, Grand Island, NY, USA) supplemented with 10% heat-inactivated fetal bovine serum, 100 units/mL penicillin and 100 µg/mL streptomycin (Life Technologies, Grand Island, NY, USA) and incubated at 37 °C in a humidified incubator with 5% CO_2_ and 95% relative humidity.

### 4.5. Immunocytochemistry

Immunocytochemical staining of FoxA1 and FoxA3 in cell lines was performed using specific primary antibodies: mouse monoclonal anti-FoxA1 (ab55178, Abcam, Cambridge, UK) or rabbit polyclonal anti-FoxA3 (sc25357, Santa Cruz Biotechnology, Santa Cruz, CA, USA) antibodies. Cell lines were placed in a 48-well plate and incubated for cell attachment. The attached cells were fixed with 10% paraformaldehyde in PBS and were then incubated with 0.2% (v/v) Triton-X100 solution for 5 min. Endogenous hydrogen peroxide activity and non-specific binding were blocked by 0.3% (v/v) hydrogen peroxide in PBS and 3% (w/v) bovine serum albumin (BSA) in PBS, respectively. The cells were then incubated with specific primary antibodies followed by peroxidase-conjugated Envision™ secondary antibodies (DAKO, Glostrup, Denmark). The signals were detected by DAB substrate kit (Vector, Laboratories, Inc., Burlingame, CA, USA). After rehydration, the cells were observed microscopically with an inverted microscope.

### 4.6. RNA Extraction and Real-Time PCR

TRIzol^®^ Reagent (Invitrogen, Carlsbad, CA, USA) was used for RNA isolation from cell pellets following the manufacturer’s protocol. The RNA quantity and quality were assessed with a NanoDrop ND-2000 spectrophotometer (NanoDrop Technologies, Wilmington, DE, USA). Then, total RNA (1.5 µg) was converted to cDNA using High-Capacity cDNA Reverse Transcription Kit (Applied Biosystems, Foster, CA, USA) according to the manufacturer’s instruction. The mRNA levels of FoxA1, FoxA3, CD133, Oct3/4 and GAPDH were detected by TaqMan gene expression assay using TaqMan probes (Hs00270129_m1 FoxA1, Hs00270130_m1 FoxA3, Hs01009257_m1 PROM1 (CD133), Hs04260367_gH POU5F1 (Oct3/4) and GAPDH). Real-time PCR was performed in an ABI real-time PCR system, Quantstudio™ 6 Flex (Life technologies, Singapore, Singapore). GAPDH was used as an internal control of the housekeeping gene.

### 4.7. FoxA1 Overexpression in CCA Cell Line

The FoxA1 expression vector (FoxA1 vector) and control vector (empty vector) were extracted from *Escherichia coli* cells containing a FoxA1 vector (pLOC-TurboRFP containing FoxA1 gene, Dharmacon Inc., Lafayette, CO, USA) or control vector (empty vector or pLOC-TurboRFP without interested element, Dharmacon Inc., Lafayette, CO, USA), respectively. The plasmid extraction was performed using GF-1 Plasmid DNA Extraction Kit (Vivantis Technologies, Selangor, Malaysia) following the manufacturer’s protocols. Cells were transiently transfected with plasmids using Lipofectamine 2000 (Invitrogen, Carlsbad, CA, USA). After 24 h of transfection, cells were harvested for RNA extraction, immunocytochemistry, cell proliferation, cell invasion and spheroid formation assays.

### 4.8. FoxA3 Knockdown Using Specific siRNA

Specific siRNA against FoxA3 (siFoxA3) (ON-TARGETplus SMARTpool Human FoxA3 siRNA, GE Healthcare Dharmacon Inc., Lafayette, CO, USA) was used for FoxA3 silencing. Scramble siRNA (ON-TARGETplus Non-targeting Control siRNA, GE Healthcare Dharmacon Inc., Lafayette, CO, USA) was used as a control siRNA (scramble). Cells were transiently transfected with siFoxA3 or scramble using Lipofectamine RNAiMAX^®^ (Invitrogen, Carlsbad, CA, USA). After 48 h of transfection, cells were harvested for RNA extraction, immunocytochemistry, cell proliferation, cell invasion and spheroid formation assays.

### 4.9. Cell Proliferation Assay

Cell proliferation was measured as cell viability using 3-(4,5-Dimethylthiazol-2-yl)-2,5-Diphenyltetrazolium Bromide (MTT) assay. The cells (3.0 × 10^4^ cells/well) were seeded in 96-well flat-bottom microtiter plate (in quintuplicate) and incubated for 24, 48 and 72 h. After incubation, the cell lines were treated with 0.5 mg/mL MTT (Sigma-Aldrich, St. Louis, MO, USA) solution in complete medium for 3 h at 37 °C. Formazan crystal was solubilized with dimethyl sulfoxide (DMSO) and measured the optical density (OD) at 540 nm using a microplate reader (Tecan Sunrise, Männedorf, Switzerland).

### 4.10. Cell Invasion Assay

Cell invasion assay was performed using Matrigel^®^ invasion chamber with 8.0-μm membrane (Corning^®^, Discovery Labware, Inc., Bedford, MA, USA). The cell lines (3.0 × 10^4^ cells/well) in serum free medium were placed into the upper chamber and the complete medium was added to the lower chamber. After 18 h of incubation, the non-invaded cells were removed, and the invaded cells were fixed with methanol and stained with Mayer’s hematoxylin. The invaded cells were reviewed and quantified under a light microscope.

### 4.11. Spheroid Formation

Ten microliters of cell suspensions (5 × 10^3^ cells) in Ham’s F-12 complete medium were hanging dropped (10 drops per condition) on the lids of cell culture dishes for 24 h at 37 °C in a humidified incubator with 5% CO_2_ and 95% relative humidity. Then, the cells were observed over an inverted microscope. This protocol was adapted from Foty R. 2011 [34].

### 4.12. Statistical Analysis

Statistical analysis was performed using SPSS Statistics software (version 19.0), IBM Cooperation, Armonk, NY, USA. The survival analysis was determined using Kaplan–Meier estimate with Log-rank test. The associations between protein expression profile and clinicopathological data were analyzed using a Pearson’s chi-square test. The t test was used for comparison of gene expression, cell proliferation, and cell invasion. The data are presented as mean ± standard deviation (SD) and the statistical significance was considered at *p* < 0.05.

## 5. Conclusions

Our results indicate that FoxA1 was down-regulated in CCA and had tumor suppressive roles in CCA progression; therefore, down-regulation of FoxA1 in CCA tissues was related with poor prognosis. FoxA2 was also down-regulated in CCA. However, the expression of FoxA2 was not correlated with any of the CCA clinical data. Therefore, FoxA2 may not play an important role in CCA progression. FoxA3 was up-regulated in CCA and had oncogenic function in CCA progression, possibly because over-expression of FoxA3 in CCA tissues was related with metastasis status. The different expression patterns and roles of FoxAs in CCA may be affected from different upstream regulators and down-stream targets. In conclusion, we found that down-regulation of FoxA1 and up-regulation of FoxA3 were involved in CCA progression. The molecular mechanisms including upstream regulators and down-stream targets of FoxAs should be highlighted in CCA progression. FoxAs, their related molecules and an in vivo model of FoxAs in CCA can be further studied for the application of CCA targeted therapy.

## Figures and Tables

**Figure 1 ijms-21-01796-f001:**
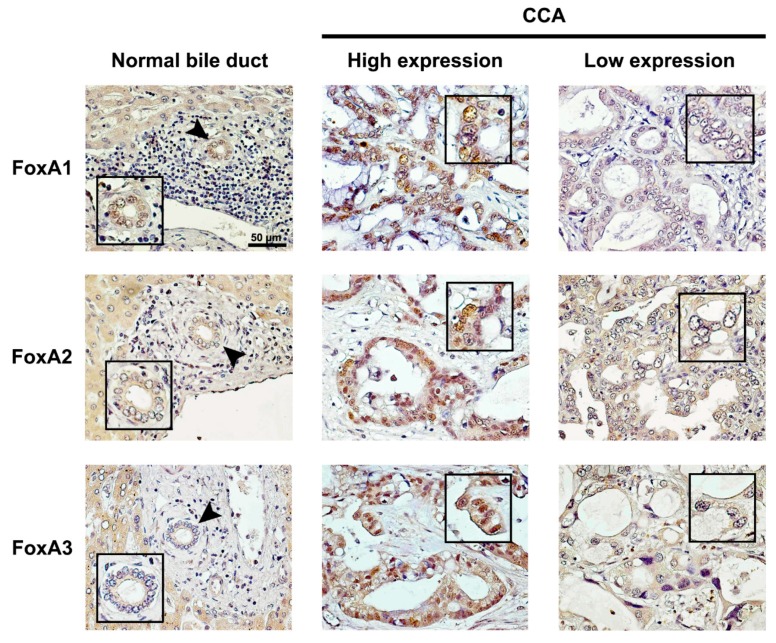
FoxA1, FoxA2 and FoxA3 expression patterns in NBD and intrahepatic CCA (high and low expression levels) analyzed by IHC. Arrow heads indicate NBD. The scale bar is equal to 50 μm.

**Figure 2 ijms-21-01796-f002:**
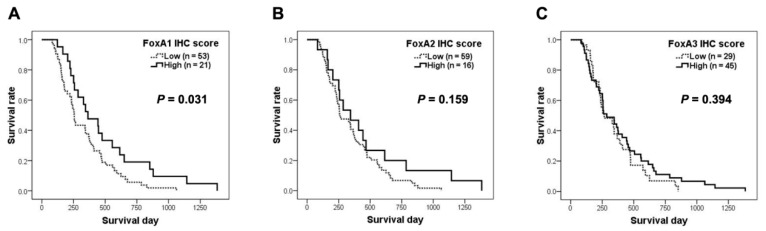
Survival analysis of FoxA1 (**A**), FoxA2 (**B**) and FoxA3 (**C**) expression patterns in intrahepatic CCA patients.

**Figure 3 ijms-21-01796-f003:**
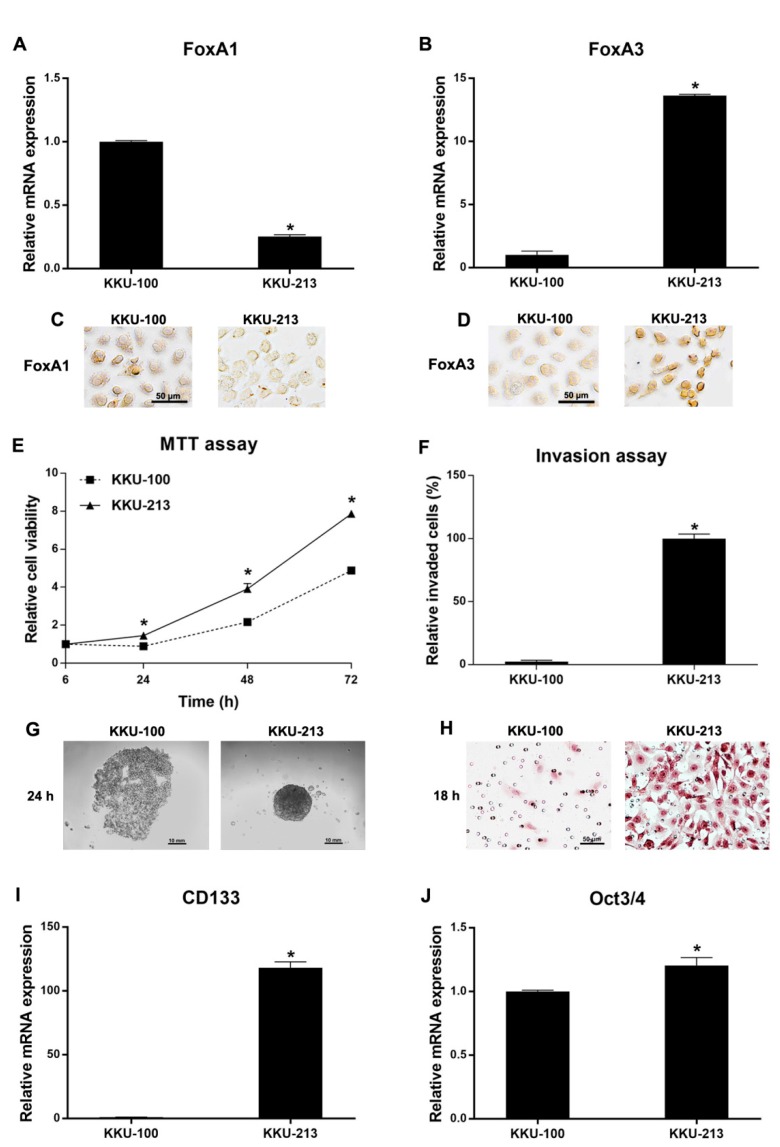
Cellular function analysis of KKU-100 and KKU-213: (**A**,**B**) mRNA expression (mean ± SD) of FoxA1 and FoxA3 measured by real-time PCR (*n* = 2 for each condition); (**C**,**D**) immunocytochemical detection of FoxA1 and FoxA3 proteins; (**E**) cell proliferation activity (mean ± SD) measured by 3-(4,5-Dimethylthiazol-2-yl)-2,5-Diphenyltetrazolium Bromide (MTT) assay (*n* = 5 per each time point per condition); (**F**) graphical representation of invaded cells (mean ± SD) in cell invasion assay (*n* = 5 per condition); (**G**) spheroid formation in hanging drop cultures; (**H**) hematoxylin-stained invaded cells under a light microscope in the cell invasion assay; and (**I**,**J**) mRNA levels (mean ± SD) of CD133 (**I**) and Oct3/4 (**J**) detected by real-time PCR (*n* = 2 per condition). * *p* < 0.05 by Student’s *t* test. All results were confirmed by three independent experiments.

**Figure 4 ijms-21-01796-f004:**
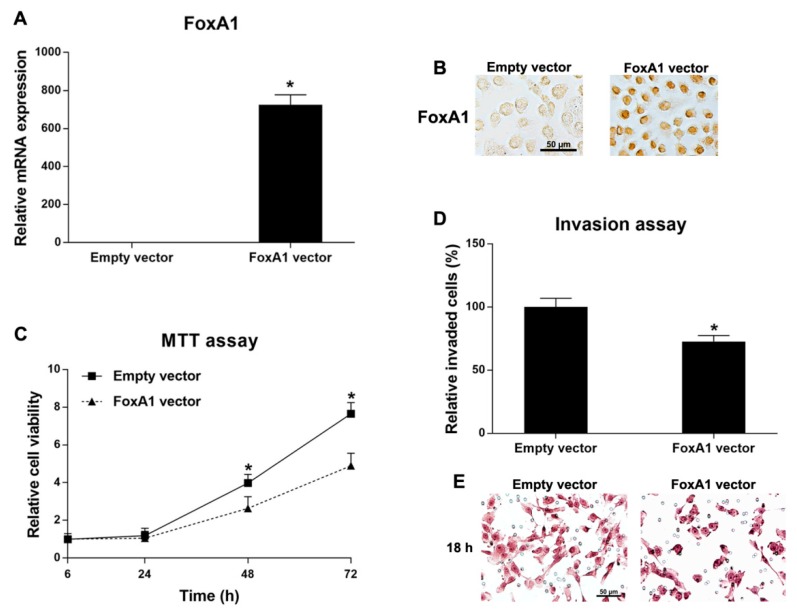
Functional analysis of FoxA1-overexpressing KKU-213 cells and the control cells: (**A**) mRNA expressions (mean ± SD) of FoxA1 measured by real-time PCR (*n* = 2 per condition); (**B**) FoxA1 protein expressions of transfected and control cells by immunocytochemical detection; (**C**) cell proliferation activity (mean ± SD) measured by MTT assay (*n* = 5 per each time point per condition); (**D**) cell invasion activity (mean ± SD) of transfected and control cells (*n* = 5 per condition); and (**E**) hematoxylin-stained invaded cells under a light microscope in the cell invasion assay. * *p* < 0.05 analyzed by Student’s *t* test. All results were confirmed by three independent experiments. Scale bars are equal to 50 μm.

**Figure 5 ijms-21-01796-f005:**
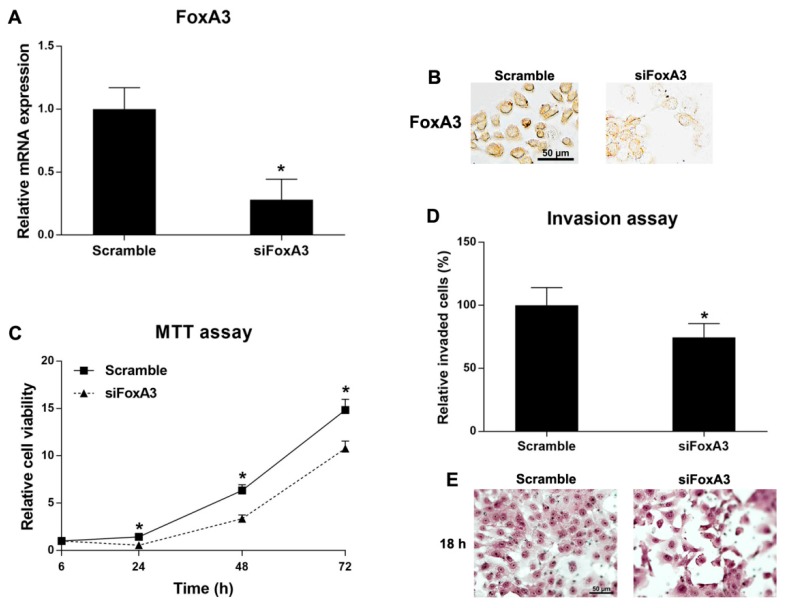
Functional analysis of FoxA3-silenced KKU-213 cells and the control cells: (**A**) mRNA expressions (mean ± SD) of FoxA3 measured by real-time PCR (*n* = 2 per condition); (**B**) FoxA3 protein expressions of the silenced and control KKU-213 cells by immunocytochemistry; (**C**) cell proliferation activity (mean ± SD) measured by MTT assay (*n* = 5 per each time point per condition); (**D**) cell invasion activity (mean ± SD) of FoxA3-silenced and control KKU-213 cells; and (**E**) hematoxylin-stained invaded cells under light microscopy in the cell invasion assay. * *p* < 0.05 analyzed by Student’s *t* test. Scale bars are equal to 50 μm. All results were confirmed by three independent experiments.

**Table 1 ijms-21-01796-t001:** Correlations of FoxA1, FoxA2 and FoxA3 expression patterns in intrahepatic CCA tissues and clinicopathological data.

Clinical Data	FoxA1			FoxA2			FoxA3		
	Low (*n* = 53)	High (*n* = 21)	*p*-value	Low (*n* = 59)	High (*n* = 15)	*p*-value	Low (*n* = 29)	High (*n* = 45)	*p*-value
**Age**									
<57	26	10	0.911 *	32	4	0.056 *	20	16	0.005 *
≥57	27	11		27	11		9	29	
**Sex**									
Male	37	14	0.792 *	41	10	0.833 *	19	32	0.612 *
Female	16	7		18	5		10	13	
**Metastasis status**									
Non-metastasis	24	9	0.850 *	27	6	0.688 *	18	15	0.015 *
Metastasis	29	12		32	9		11	30	
Median survival (days)	256	364	0.031 ^#^	260	344	0.159 ^#^	266	286	0.394 ^#^

* *p*-value was analyzed by Pearson’s Chi-square test. ^#^
*p*-value was analyzed by log-rank test.

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
