# Peer review of "Opposing Roles of FoxA1 and FoxA3 in Intrahepatic Cholangiocarcinoma Progression"

_ijms, 2020, doi:10.3390/ijms21051796_

Round 1
Reviewer 1 Report
General comment
The authors attend to investigate the role of FOXAs proteins in the context of human cholangiocarninoma. Analysis of human liver by immunohistochemistry quantification, survival analysis, and in vitro experiments with cell lines are performed. Although conceptually interesting, the study lacks precision, details, and rigorousness. No in vivo model is provided demonstrating the key role of these proteins in CCA. This manuscript would need important modifications and improvement to reach standard for publications in IJMS.
Specific comments
Figure 1
Please provide isotype controls and dot plots with statistics summarizing all the data. (how many patients and location per slide ? , paired samples inside outside the tumor ?)
Please indicate the scale on the picture.
Each time the author refer to CCA, please indicate which type (intrahepatic ?)
Some zoom might be provided to better interpret and distinguish the stainings patterns
Table 1
Please as a validation cohort, provide data from one or two datasets such a TCGA for example.
An analysis pan-cancer could be provided in supplement to evaluate the relevance of these proteins in other cancers.
Please perform a multivariate cox regression model and provide the results.
From public datasets analysis, what genes transcripts correlate the most with the expression of these proteins? That would help to gain insight into their specific regulatory modules.
Figure 3
Please provide another cell line as a negative control as well as isotype controls
Please indicate the interquartile range on the graphics, or the SD if the data are normally distributed.
For all graphs, please indicate the number of experiments and replicates.
If possible please test the hypothesis with an in vivo model of CCA.
Author Response
Reviewer 1
The authors attend to investigate the role of FOXAs proteins in the context of human cholangiocarninoma. Analysis of human liver by immunohistochemistry quantification, survival analysis, and in vitro experiments with cell lines are performed. Although conceptually interesting, the study lacks precision, details, and rigorousness. No in vivo model is provided demonstrating the key role of these proteins in CCA. This manuscript would need important modifications and improvement to reach standard for publications in IJMS.
Specific comments
Comment#1: Figure 1, please provide isotype controls and dot plots with statistics summarizing all the data. (how many patients and location per slide ? , paired samples inside outside the tumor ?)
Reply to comment#1
In the IHC staining, negative controls were performed by omitting each primary antibody which were performed at the same time of the experiments that showed in Figure 1 (revised manuscript). The staining results, which are shown in Supplementary Figure R1, indicated that there was no cross reactivity between each secondary antibody and the protein of interest.
Please see Figure R1 in the attached file: Negative controls of FoxA1, FoxA2 and FoxA3 staining in CCA tissues using immunohistochemistry. Scale bar equals 20 mm.
We obtained the CCA paraffin sections from 74 patients from the specimen bank of Cholangiocarcinoma Research Institute, Khon Kaen University. Among 74 specimens, 39 had a pair of cancerous (CCA) and adjacent non-cancerous areas (containing normal bile ducts; NBD). The dot plots of FoxA1, FoxA2 and FoxA3 expressions in noncancerous and cancerous areas of 39 CCA tissues were shown in Figure R2.
Please see Figure R2 in the attached file: Dot plots and the mean ± SD of IHC scores of FoxA1 (A), FoxA2 (B) and FoxA3 (C) expressions in normal bile duct (NBD) and cancer (CCA) areas of CCA tissues. The FoxA1 expression level in CCA cancer areas was slightly lower than that in NBD cells (p = 0.048, analyzed by Wilcoxon Signed Ranks Test). The FoxA2 expression level in the cancer cells was also significantly lower than that in NBD cells (p < 0.001). In contrast, the FoxA3 expression in the cancer cells was significantly higher than that in the NBD cells (p < 0.001).
We did not include the aforementioned results in the revised manuscript because of insufficient sample number of the paired specimens of cancerous and adjacent noncancerous tissues.
Comment#2: Figure1, please indicate the scale on the picture.
Reply to comment#2:
A scale bar has been added in the Figure 1 (FoxA1 expression in normal tissues). All photographs are taken at the same magnification.
Comment#3: Each time the author refer to CCA, please indicate which type (intrahepatic ?)
Reply to comment#3:
All CCA tissues used in this study were of the intrahepatic type. This point has been added to the revised manuscript.
Comment#4: Figure 1, some zoom might be provided to better interpret and distinguish the stainings patterns
Reply to comment#4:
All photographs in the revised Fig. 1 have been enlarged.
Comment#5: Table 1, please as a validation cohort, provide data from one or two datasets such a TCGA for example. An analysis pan-cancer could be provided in supplement to evaluate the relevance of these proteins in other cancers.
Reply to comment#5:
-Thank you for your suggestion. The TCGA pan-cancer analyze is very interesting. We have added the expression data across 33 different cancer types using TCGA database to the manuscript. The data provided clearer information of the FoxAs in other cancers. TCGA pan-cancer analyses of FoxA1, FoxA2 and FoxA3 were successfully performed (Figure S1A, S1B and S1C).
Moreover, using the Human Protein Atlas database (www.proteinatlas.org) we identified the expressions of FoxAs in different cancer types (i.e. glioma, thyroid cancer, lung cancer, colorectal cancer, head and neck cancer, stomach cancer, liver cancer, carcinoid, pancreatic cancer, renal cancer, urothelial cancer, prostate cancer, testis cancer, breast cancer, cervical cancer, endometrial cancer, ovarian cancer, melanoma, skin cancer and lymphoma).
FoxA1; https://www.proteinatlas.org/ENSG00000129514-FOXA1/pathology
FoxA2; https://www.proteinatlas.org/ENSG00000125798-FOXA2/pathology
FoxA3; https://www.proteinatlas.org/ENSG00000170608-FOXA3/pathology
Data obtained from the Human Protein Atlas database were summarized as follows: FoxA1 was highly expressed in breast, prostate, urothelial cancers and carcinoids. Its expression levels were low in lung, cervical and endometrial cancers. Other cancer tissues were in general weakly stained or negative. Low expression of FoxA1 in urothelial cancer was correlated with poor prognosis that was similar to our finding in CCA patients.
FoxA2 was highly expressed in breast, prostate, urothelial, gastric, hepatocellular and gynaecological cancers. Other malignancies were generally negative. Low FoxA2 expression correlated with poor prognosis in endometrial, pancreatic and ovarian cancers. In this study, most of CCA tissues showed low FoxA2 expression and there was no correlation between FoxA2 expression levels and the patient prognosis.
FoxA3 was highly expressed in carcinoid and glioma but most of other malignancies showed weak to moderate expressions. No information is available regarding FoxA3 expression levels and other clinical data on the website. In CCA, high FoxA3 expression level was significantly correlated with positive metastasis status and ages of CCA patients.
Nevertheless, it is rather beyond the scope of this study. We therefore did not include the above described data in the Discussion section.
Comment#6: Table 1, please perform a multivariate cox regression model and provide the results.
Reply to comment#6:
Thank you for your advice. We have performed a Cox proportional-hazards model and the results showed that the FoxA1 expression level was an independent factor that for the overall survival in CCA patients as shown in Table R1. Accordingly, we added “Moreover, the Cox proportional-hazards model indicated that the FoxA1 expression level was an independent factor for the overall survival in CCA patients (p = 0.031, adjusted hazard ratio = 1:0.558 and 95% CI = 0.326-0.953)” in Topic 2.2 (lines 106-108 of the revised manuscript).
Table R1 Multivariate cox regression analysis of FoxA1 expression and overall survival rates of CCA patients
|
Factor |
Adjusted hazard ratio |
95% CI |
P value |
|
FoxA1 expression levels Low High |
1 0.558 |
0.326-0.953 |
0.031 |
Comment#7: From public datasets analysis, what genes transcripts correlate the most with the expression of these proteins? That would help to gain insight into their specific regulatory modules.
Reply to comment#7:
From Reactome pathway analysis (www.reactome.org) and the literature, FoxA1 is related to the expression of estrogen responsive genes such as TFF1 and BIRC5. We performed three independent experiments to measure the mRNA expression levels of the TFF1 and BIRC5 genes in FoxA1-overexpressing KKU213 and the control cells. However, the results obtained were controversial or inconsistent; we, therefore, could not interpret the results.
Besides, from the Reactome pathway analysis and the literature, FoxA3 is correlated with FoxA1, FoxA2 and lipid metabolism. We performed there independent experiments to measure the mRNA and protein expression levels of FoxA1 and FoxA2 in FoxA3-suppressing-KKU213 and the control cells. However, the suppression of the FoxA3 did not affect the expression levels of FoxA1 and FoxA2. In other words, FoxA3 had no direct effect on FoxA1 and FoxA2 expressions in CCA cells. The above data have been added at the end of Discussion section.
Comment#8: Figure 3, please provide another cell line as a negative control as well as isotype controls
Reply to comment#8:
We screened the mRNA expressions of FoxA1 and FoxA3 in an immortal cholangiocyte cell line (MMNK1) and four different CCA cell lines (i.e., KKU-213, KKU-214, KKU-156 and KKU-100) as shown in Fig. R3A and R3B. The MMNK1 cell line had low expression levels of FoxA1 and FoxA3. The overall expression level of FoxA1 was significantly higher in all CCA cell lines in order from the greatest to least: KKU-100 > KKU-214 > KKU-156 > KKU-213. The expression level of FoxA3 was significantly higher in three CCA cell lines in order from the greatest to least: KKU-214 > KKU-156 > KKU-213.
Among those cell lines, KKU-213 and KKU-100 were selected for the comparison of tumorigenic properties because they have cell line certificates from the Japanese Collection of Research Bioresources (JCRB) Cell Bank (JCRB1568; KKU-100 and JCRB1557; KKU-213). This allows researchers to have an easy access to the available information on the above mentioned cell lines as well as to perform reproducible experiments.
Please see Figure R3 in the attached file: Relative mRNA expression levels of FoxA1 (A) and FoxA3 (B) in an immortal cholangiocyte cell line (MMNK1) and four different CCA cell lines (KKU-213, KKU-214, KKU-156 and KKU-100).
- In the case of the immunocytochemistry staining, negative controls were run omitting primary antibodies. All negative controls did not stained brown (no positive staining of DAB).
Comment#9: Figure 3, please indicate the interquartile range on the graphics, or the SD if the data are normally distributed.
Reply to comment#9:
All data are presented as mean ± SD. SD were represented by error bars.
Comment#10: For all graphs, please indicate the number of experiments and replicates.
Reply to comment#10:
All results showed in this manuscript were performed as three independent experiments. The experiments and replicates were added in the figure legends.
Comment#11: If possible please test the hypothesis with an in vivo model of CCA.
Reply to comment#11:
An animal ethics approval is required for every in vivo study. Unfortunately, since the beginning, as we have not planned to perform any in vivo experiment, we did not have any animal ethics approval for that. Certainly, performing an in vivo experiment could be considered in our future work.

Reviewer 2 Report
The manuscript by Thanan et al. describes the role of FoxA1 and FoxA3 genes inCCA progression.
The authors suggest an opposing role of the 2 above-mentioned genes based on their different expression levels in CCA tissues from affected patients and assessed in cell lines their hypothesis supporting their original findings.
English style has to be checked throughout the text; especially in the introduction where there are many misleading sentences.
Limits of the study should be also assessed in the discussion or conclusions.
Author Response
Reviwer#2
Comment#1: The manuscript by Thanan et al. describes the role of FoxA1 and FoxA3 genes in CCA progression. The authors suggest an opposing role of the 2 above-mentioned genes based on their different expression levels in CCA tissues from affected patients and assessed in cell lines their hypothesis supporting their original findings.
Reply to comment#1:
Thank you very much for your supportive comments.
Comment#2: English style has to be checked throughout the text; especially in the introduction where there are many misleading sentences.
Reply to comment#2:
We have carefully revised the manuscript, especially in the Introduction section. We truly believe that this revised manuscript is significantly improved and suitable for publication in IJMS.
Comment#3: Limits of the study should be also assessed in the discussion or conclusions.
Reply to comment#3:
The limitations of this study are described as follows: 1) supportive evidence of molecular mechanisms underlying FoxA3 induced carcinogenesis is lacking and 2) in vivo experimental data on the roles of FoxA1/FoxA3 in CCA progression are presently unavailable. We have already added the limitations in the Conclusion section of the revised manuscript.
Round 2
Reviewer 1 Report
The authors addressed all comments and modified the manuscript accordingly,
No additional comments is made from my side.